# Dendritic Resonate-and-Fire Neuron for Effective and Efficient Long Sequence Modeling

**Dehao Zhang**[1], **Malu Zhang**[1,2]*, **Shuai Wang**[1], **Jingya Wang**[1], **Wenjie Wei**[1], **Zeyu Ma**[1],
**Guoqing Wang**[1], **Yang Yang**[1], **Haizhou Li**[2,3]

[1]University of Electronic Science and Technology of China
[2]Shenzhen Loop Area Institute, [3]The Chinese University of Hong Kong (Shenzhen)

## Abstract

The explosive growth in sequence length has intensified the demand for effective and efficient long sequence modeling. Benefiting from intrinsic oscillatory membrane dynamics, Resonate-and-Fire (RF) neurons can efficiently extract frequency components from input signals and encode them into spatiotemporal spike trains, making them well-suited for long sequence modeling. However, RF neurons exhibit limited effective memory capacity and a trade-off between energy efficiency and training speed on complex temporal tasks. Inspired by the dendritic structure of biological neurons, we propose a Dendritic Resonate-and-Fire (D-RF) model, which explicitly incorporates a multi-dendritic and soma architecture. Each dendritic branch encodes specific frequency bands by utilizing the intrinsic oscillatory dynamics of RF neurons, thereby collectively achieving comprehensive frequency representation. Furthermore, we introduce an adaptive threshold mechanism into the soma structure. This mechanism adjusts the firing threshold according to historical spiking activity, thereby reducing redundant spikes while maintaining training efficiency in long-sequence tasks. Extensive experiments demonstrate that our method maintains competitive accuracy while substantially ensuring sparse spikes without compromising computational efficiency during training. These results underscore its potential as an effective and efficient solution for long sequence modeling on edge platforms.

## 1 Introduction

Long sequence modeling efficiently captures complex temporal patterns and dynamic characteristics, demonstrating exceptional application potential in edge computing scenarios such as speech recognition [35, 64, 65] and electroencephalogram (EEG) monitoring [5, 48, 67]. However, mainstream sequence modeling methods still primarily rely on Recurrent Neural Networks (RNNs) [49, 52], Transformers [29, 62], and state-space models (SSMs) [18, 20]. Although these approaches effectively compress contextual information into finite states, they still involve extensive floating-point matrix multiplications, resulting in high computational complexity, inference latency, and energy consumption [51]. Therefore, designing long sequence models that simultaneously achieve high performance, energy efficiency, and fast inference remains an essential and ongoing research challenge.

Inspired by the structure and function of neural circuits in the brain, Spiking Neural Networks (SNNs) have emerged as a biologically plausible and computationally efficient model [37, 59]. Unlike Artificial Neural Networks (ANNs), SNNs possess event-driven computational capabilities [6, 77, 78] and the potential to process dynamic temporal information [72, 70, 80]. These properties allow information to be transmitted through binary spikes and enable the retention of historical context

---

*Corresponding author: maluzhang@uestc.edu.cn

via membrane potentials. Several studies have applied SNNs for long sequence modeling based on Leaky Integrate-and-Fire (LIF) neurons. Zhang et al. [79] introduce a two-compartment LIF neuron to better capture long-term temporal dependencies, while Fang et al. [14] remove the reset mechanism to maximize the utilization of information across all timesteps. However, due to their serial charge-fire-reset dynamics, these methods fail to capture complex timescale variations and long-range dependencies in long sequences [66, 72, 76], limiting their performance in complex tasks.

As an efficient alternative to LIF neurons, Resonate-and-Fire (RF) neurons [27] employ complex-valued state variables to more effectively retain historical information in long sequences modeling. Higuchi et al. [21] integrate a refractory mechanism into the reset dynamics of RF neurons, maintaining frequency selectivity while significantly improving long sequence processing capacity and promoting spike pattern sparsity. Huang et al. [24] remove the reset mechanism and propose that a learnable time constant can capture the intrinsic reset behavior of RF neurons. This approach reduces the computational complexity from $\mathcal{O}(\mathcal{L}^2)$ to $\mathcal{O}(\mathcal{L} \log \mathcal{L})$, thereby substantially improving computational efficiency. Despite these advancements, current methods still face two major challenges. First, the limited bandwidth of RF neurons constrains their ability to extract diverse frequency-band combinations from complex temporal signals, making them behave more like simplified resonators. Second, RF neurons encounter a trade-off between energy efficiency and training speed: removing the reset mechanism enables efficient training but leads to excessive spike activity, whereas incorporating a reset suppresses spiking activity at the cost of increased training overhead.

Inspired by the dendritic structure of biological neurons [44, 61, 38, 26], we propose a novel Dendritic Resonate-and-fire (D-RF) neuron for effective and efficient long sequence modeling. It consists of a multi-dendrite and soma structure. First, each dendritic branch captures specific frequency responses from input signals through the oscillation characteristics of RF neurons, thereby achieving comprehensive spectral decomposition across multiple timescales. Second, we incorporate an adaptive threshold mechanism into the soma structure that dynamically adjusts thresholds based on historical spiking patterns, achieving sparse spikes while maintaining training efficiency. Extensive experiments on long sequence tasks confirm the high-performance and energy-efficient of the proposed neuron model. The main contributions are summarized as follows:

- We conduct a detailed analysis of the limitations of existing RF neurons in long sequence modeling, highlighting their restricted memory capacity and the inherent trade-off between energy efficiency and training speed. First, the limited bandwidth response causes RF neurons to behave like simplified resonators. Second, the presence or absence of a reset mechanism leads to a conflict between sparse spiking and training efficiency.

- We propose the D-RF neuron, which comprises two components. First, dendritic branches exploit RF dynamics to achieve specialized frequency selectivity, collectively enabling full spectral coverage across multiple timescales. Second, an adaptive threshold mechanism in the soma dynamically adjusts thresholds based on history spiking activity, balancing computational cost and energy efficiency while preserving training effectiveness.

- Extensive experiments demonstrate that our method achieves competitive performance across various long sequence tasks. Moreover, the method produces sparser spiking activity while maintaining training efficiency. These results highlight our model's dual advantages in effectiveness and computational efficiency for long sequence modeling.

## 2 Related Work

### 2.1 Advanced Spiking Neurons for Long Sequence Modeling

Due to the dynamic characteristics of spiking neurons, it is believed that they have the ability to handle long sequence modeling. However, the LIF model [17, 28] and its variants [4, 13, 72] exhibit limited memory capacity in temporal tasks, which is considered a critical factor for effective long sequence modeling. To overcome this limitation, several studies [7, 79] draw inspiration from more complex neural dynamics. RF neurons [27] have attracted considerable attention due to their intrinsic frequency band preference. Orchard et al. [41] leverage RF neurons to encode raw signals into sparse spike trains, thereby significantly reducing output bandwidth. Furthermore, Higuchi et al. [21] introduce adaptive decay factor mechanisms [50, 16] and refractory mechanisms [54], improving the balance between energy efficiency and performance of RF neurons in long-range sequence modeling. In addition,

RF-based models demonstrate competitive performance in sequence modeling tasks, including image classification [22], optical flow tracking [15], and audio processing [53, 75]. Moreover, RF neurons can be efficiently implemented on neuromorphic hardware like the Loihi [9, 41].

## 2.2 Training Strategies for Spiking Neural Networks

The mainstream training methods for deep SNNs can be categorized into ANN-to-SNN conversion [12, 47, 63] and direct training [68, 69]. ANN-to-SNN conversion methods use the similarity between spike firing rates and ANN activation functions, but need many timesteps to reach high accuracy. In contrast, direct training enables SNNs to achieve performance comparable to ANNs with the same architecture within a limited number of timesteps. Specifically, direct training introduces surrogate gradient functions [11, 40] to enable backpropagation, thereby addressing the non-differentiability of spike firing functions. However, applying direct training to long sequence tasks presents greater challenges [51], as such tasks often require thousands of timesteps. Consequently, some research [23, 58, 76] tend to explore more efficient training strategies for SNNs. Yin et al. [73] further explore intrinsic challenges in SNNs training, especially the reset behavior of neurons. Therefore, Fang et al. [14] change the dynamic process of spiking neurons into a learnable matrix, avoiding the reset mechanism. Similarly, Shen et al. [51] introduce the SDN block to simulate the reset process. These methods greatly reduce the training cost of SNNs while keeping their asynchronous inference ability.

# 3 Preliminary

## 3.1 Resonate-and-Fire Neuron

Inspired by the damped and sustained subthreshold oscillations observed in the membrane potentials of mammalian nervous systems [2, 34, 42, 46], RF neurons are proposed [27]. Given input signal $\mathcal{I}(t)$, the dynamics of the RF neuron at timestep $t$ can be described as follows:

$$\frac{d}{dt}z(t) = (b + \mathrm{i}\omega)z(t) + \mathcal{I}(t), \tag{1}$$

$z = u + \mathrm{i}v \in \mathbb{C}$ represents the complex state of RF neurons, where $u$ represents a current-like variable capturing voltage-gated and synaptic current dynamics, and the imaginary component $v$ corresponds to a voltage-like variable. $\omega > 0$ denotes the angular frequency of the neuron, indicating the number of radians it oscillates per second, while the damping factor $b < 0$ regulates the exponential decay of the oscillation. It can be discretized using the Euler method [3]:

$$z[t] = \exp\{\delta(b + \mathrm{i}\omega)\} \cdot z[t-1] + \delta\mathcal{I}[t], \tag{2}$$

$\delta$ is the discrete timestep. When the real part of $z[t]$ exceeds the threshold, the neuron fires a spike; otherwise, it remains silent. Additionally, RF neuron exhibits a preference for specific frequency bands. As shown in Fig. 1(a), we present the oscillatory behavior under spike inputs with different frequencies. It is observed that both the membrane potential and the phase state accumulate rapidly.

## 3.2 Direct Training in Spiking Neural Networks

Due to the BPTT [68] and Surrogate Gradient methods [40], training large-scale SNNs becomes feasible. Specifically, the gradient of the weight $w^l$ at timestep $T$ can be represented as follows:

$$\nabla_{w^l}\mathcal{L} = \sum_{l=1}^{T}\left(\frac{\partial\mathcal{L}}{\partial u^l(t)}\right)S^{l-1}(t), \tag{3}$$

where $u^l(t)$ and $S^l(t)$ denote the membrane potential and spike emission of the $l$-th layer at time $t$, respectively. $\mathcal{L}$ is the loss function. It can be calculated as follows:

$$\frac{\partial\mathcal{L}}{\partial u^l(t)} = \frac{\partial\mathcal{L}}{\partial S^l(t)}\frac{\partial S^l(t)}{\partial u^l(t)} + \frac{\partial\mathcal{L}}{\partial u^l(t+1)}\frac{\partial u^l(t+1)}{\partial u^l(t)}, \tag{4}$$

$$\frac{\partial\mathcal{L}}{\partial S^l(t)} = \frac{\partial\mathcal{L}}{\partial u^{l+1}(t)}\frac{\partial u^{l+1}(t)}{\partial S^l(t)} + \frac{\partial\mathcal{L}}{\partial u^l(t+1)}\frac{\partial u^l(t+1)}{\partial S^l(t)}. \tag{5}$$

The non-differentiable term $\frac{\partial S^l(t)}{\partial u^l(t)}$ can be substituted with a surrogate function. $\frac{\partial u^l(t+1)}{\partial S^l(t)}$ and $\frac{\partial u^l(t+1)}{\partial u^l(t)}$ are the temporal gradients that need to be calculated. As shown in Eq. 4 and Eq. 5, the gradient at each timestep depends not only on the current states but also recursively on future states. This recursive dependency across layers and timesteps results in a computational complexity of $\mathcal{O}(\mathcal{L}^2)$.

## 4 Method

### 4.1 Problem Analysis

**Limited Performance in Complex Tasks**: Benefiting from the design of the decay kernel and complex-valued states, RF neurons exhibit pronounced frequency selectivity. However, this property also limits the model's ability to discriminate diverse input patterns. As shown in Fig. 1(b), input signals with distinct frequency components may elicit similar spiking responses, as components misaligned with the neuron's intrinsic frequency tend to be suppressed, thereby impairing the network's capacity to capture and distinguish diverse temporal features. To further verify this limitation, we visualize the frequency response of a single RF neuron. The results show that the neuron responds primarily within a narrow bandwidth and reaches its peak at the intrinsic frequency, making it difficult to capture complex frequency compositions. This observation highlights an inherent representational limitation of RF neurons in modeling complex temporal features.

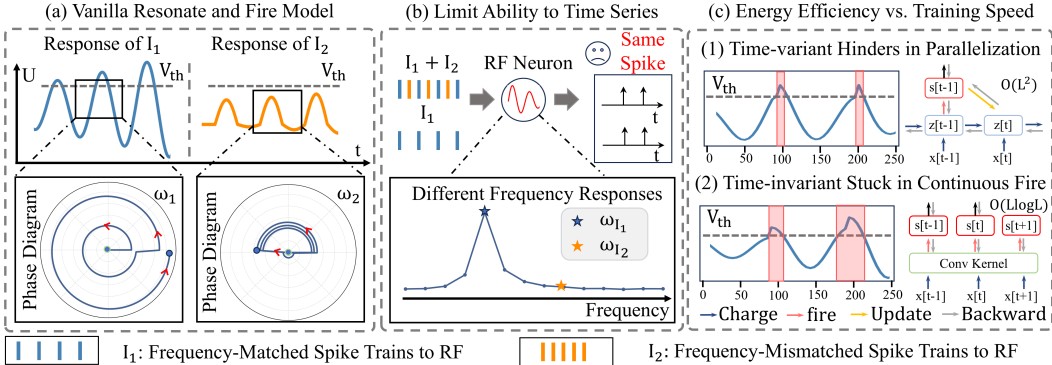

Figure 1: Problem Analysis: (a) Response of Different Spike Trains: Frequency-matched inputs lead to rapid membrane potential accumulation, whereas mismatched inputs yield weaker responses. (b) Limited Ability to Time Series: A single RF neuron struggles to respond to different frequency-varying inputs due to its narrow band selectivity. (c) Energy Efficiency vs. Training Speed: The time-variant method enables sparse spiking but has a high training cost of $\mathcal{O}(\mathcal{L}^2)$. The time-invariant method allows faster training with complexity of $\mathcal{O}(\mathcal{L} \log \mathcal{L})$, but often leads to continuous fire.

**Challenges in Balancing Energy Efficiency and Training Speed**: In long sequence tasks, RF neurons incur significant training costs and potentially excessive energy consumption. Existing research focuses on two main strategies. First, as shown in Fig. 1(c).1, Higuchi et al. [21] propose an adaptive decay factor that instantly raises the decay coefficient after each spike, suppressing membrane potential oscillations and promoting sparse spikes. However, this approach requires temporal unfolding during backpropagation, resulting in $\mathcal{O}(\mathcal{L}^2)$ complexity. Second, Huang et al. [24] employ a learnable decay factor to the reset mechanism and use a Fourier transform reformulation to reformulate the temporal dynamics into a parallel convolutional process between an oscillatory kernel and the input signal, reducing computational complexity to $\mathcal{O}(\mathcal{L} \log \mathcal{L})$. Nevertheless, due to the decay factor remains time-invariant, the model preserves its pre-spike amplitude. As shown in Fig. 1(c).2, it results in sustained burst firing that undermines the low-power advantages of SNNs.

### 4.2 Dendritic Resonate-and-Fire Neuron

To better capture features across different frequency bands, we propose the D-RF neuron model. Unlike the vanilla RF model [27], D-RF model comprises a soma and multiple dendritic branches. Each dendritic branch extracts state responses corresponding to specific frequency preferences in the input signal $\mathcal{I}[t]$. When $\mathcal{I}[t]$ contains frequency components aligned with a branch's preference, the

membrane potential of that branch accumulates rapidly. The soma integrates input currents from all dendritic branches and generates a spike once its membrane potential exceeds a predefined threshold. Specifically, the membrane potential dynamics of the $i$-th dendritic branch is defined as follows:

$$\frac{dz_i(t)}{dt} = \{-1/\tau_i + i\omega_i\} \cdot z_i(t) + \gamma_i \mathcal{I}(t), \tag{6}$$

where $\tau_i$ and $\omega_i$ represent the decay factor and membrane potential oscillation coefficient associated with the $i$-th dendritic branch, respectively. $\mathcal{I}(t)$ denotes the presynaptic input at time $t$, and $\gamma_i$ represents the membrane capacitance of the $i$-th dendritic branch. This modeling framework allows different dendritic branches to selectively respond to specific frequency components. To enable efficient inference, the Zero-Order Hold (ZOH) method [10] is employed for discretization. The membrane potential dynamics of all dendrites are given by:

$$\mathcal{Z}[t] = \exp\left\{\begin{bmatrix} -\frac{1}{\tau_1} + i\omega_1 & 0 & \cdots & 0 \\ 0 & -\frac{1}{\tau_2} + i\omega_2 & \cdots & 0 \\ \vdots & \vdots & \ddots & \vdots \\ 0 & 0 & \cdots & -\frac{1}{\tau_n} + i\omega_n \end{bmatrix} \cdot \delta \right\} \mathcal{Z}[t-1] + \Gamma^l I[t]. \tag{7}$$

Here, $\delta$ denotes the discrete timestep, and $\mathcal{Z} = [z_1, z_2, \cdots, z_n]^T$ represents the states of individual dendritic branches, with $\Gamma = [\gamma_1, \gamma_2, \cdots, \gamma_n]^T$ denoting their respective time constants. To further enhance the frequency characteristics across dendritic branches, each branch is assigned an individual importance weight. The dynamics of soma are defined as follows:

$$H[t] = \mathcal{C}\Re\{Z(t)\}, \qquad S[t] = \Theta\left(H[t] - V_{th}[t]\right), \tag{8}$$

$\mathcal{C} \in \mathbb{R}^{n \times 1}$ denotes the importance weights assigned to each dendritic branch. $\Theta(\cdot)$ represents the Heaviside function. When the presynaptic membrane potential of the soma $H[t]$ exceeds the threshold $V_{th}$, a spike $S[t]$ is generated.

We further analyze the effectiveness of the dendrite design by examining its frequency band responses. For time-invariant RF neuron with decay kernel $b + i\omega$, it can be modeled as a time-invariant convolutional process with kernel $h(n)$, defined as $h(n) = \exp\{\delta(b + i\omega)\}^n$. Detailed proof is provided in the Appendix. A. Its frequency response is described as follows:

$$||H(\exp\{i\Omega\})|| = \left\|\sum_{n=0}^{\infty} h(n)\exp\{-i\Omega n\}\right\| = \left\|\frac{\delta}{1 - \exp\{\delta b + i(\delta\omega - \Omega)\}}\right\|. \tag{9}$$

Therefore, a single RF neuron can be regarded as a first-order band-pass filter with a resonance peak at $\Omega \approx \omega$, and a narrow frequency band determined by the damping factor $\delta b$ (as shown in Fig. 1(b)). In contrast, the frequency response of our D-RF model is defined as follows:

$$||H_{\text{D-RF}}(\exp\{i\Omega\})|| = \sum_{i=1}^{n} \mathcal{C}_i \cdot ||H(\exp\{i\Omega\})||, \quad B_{\text{eff}} \approx \sum_{i=1}^{n} \beta_i \left\|\frac{\tau_i}{\delta}\right\|, \tag{10}$$

$B_{\text{eff}}$ denotes the total frequency response of the D-RF neuron, while $\beta_i \in [0, 1)$ quantifies the independent contribution of the $i$-th branch to the overall frequency coverage. Consequently, compared to its single-branch counterpart, the D-RF neuron provides a significantly broader frequency sensitivity.

### 4.3 Adaptive Threshold for Accelerated and Efficient Learning

To balance training speed and energy efficiency, we propose an adaptive thresholding strategy that dynamically adjusts the threshold based on the spiking activity from previous timesteps. Specifically, the threshold at timestep $t$ is defined as:

$$V_{th}[t] = \sum_{k=1}^{n} \alpha_k \Theta(\Re\{\mathcal{Z}[t-k-1, ..., t-1]\} - V_{\text{pre}}) + V_{\text{pre}}, \tag{11}$$

where $V_{\text{pre}}$ denotes the origin threshold (set to 1), and $\alpha_k \in (0, 1)$ represents the importance of preceding spikes. As shown in Fig. 2(b), $\alpha_k$ is the shared parameter during the adaptive threshold

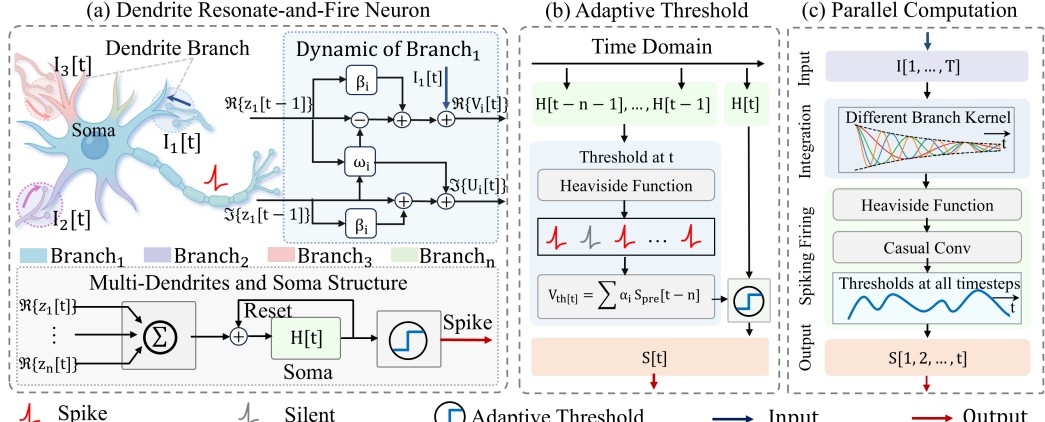

(a) Dendrite Resonate-and-Fire Neuron     (b) Adaptive Threshold     (c) Parallel Computation

Figure 2: The Structure of D-RF model: (a) Dendrite Resonate-and-Fire Neuron: D-RF model consists of multiple dendritic branches, each encoding frequency-specific dynamics. The soma integrates membrane potentials from all branches through an adaptive threshold mechanism to enable sparse spikes. (b) Adaptive Threshold: The threshold at time t is dynamically determined based on the historical spiking activity. (c) Parallel Computation: Different convolution kernels on each branch and a causal convolution for the adaptive threshold allow parallel processing of the input over time.

computation. This process can be interpreted as a one-dimensional causal convolution with kernel size $n$, where the kernel is defined as $\mathcal{A} = [\alpha_1, \ldots, \alpha_n]$. The spiking process can be reformulated as:

$$\mathcal{S}[t] = \Theta\{\underbrace{C^l\Re\{\mathcal{Z}\}}_{\text{Dendritic Input}} - \underbrace{(\text{Conv1d}(\Theta(C^l\Re\{\mathcal{Z}\} - V_{\text{pre}}) + V_{\text{pre}})}_{\text{Adaptive Threshold}}\}[t]. \quad (12)$$

Conv1d$(\cdot)$ present causal convolution along the temporal domain, enabling sparser spike activity while preserving the parallelizable nature of training. We demonstrate the effectiveness of this strategy by analyzing both the forward and backward propagation processes.

In the forward propagation, the computational complexity of the D-RF neuron is determined by multi-dendritic input and an adaptive threshold mechanism. Without the need for temporal unfolding, the D-RF neuron can be formulated in a parallelized manner. For the component of dendritic input, each dendritic branch functions independently and is decoupled in time. Therefore, the membrane potentials of all branches at time t are defined as follows:

$$\mathcal{Z}^l[t] = \sum_{k=0}^{t} \Gamma^l \exp\{k \cdot \delta\mathcal{D}\} \cdot \mathcal{I}^l[t-k], \quad (13)$$

$\mathcal{D}$ characterizes the oscillatory resonators of individual dendritic branches and is defined as a diagonal matrix: $\mathcal{D} = \text{Diag}\{-1/\tau_1 + \omega_1, \ -1/\tau_2 + \omega_2, \ \cdots, \ -1/\tau_n + \omega_n\}$. Since $\mathcal{D}$ is time-invariant, Eq. 13 can be reformulated as a convolution between the input signal $\mathcal{I}$ and the kernel $\mathcal{K}$:

$$\mathcal{Z}[t] = (\mathcal{K} * \mathcal{I})[t] = \mathcal{F}^{-1}\{\mathcal{F}\{\mathcal{K}\} \cdot \mathcal{F}\{\mathcal{I}\}\}[t], \quad \mathcal{K} = [\delta\mathcal{D}^1, \ \delta\mathcal{D}^2, \ \ldots, \ \delta\mathcal{D}^n], \quad (14)$$

$\mathcal{F}(\cdot)$ and $\mathcal{F}^{-1}(\cdot)$ denote the forward and inverse Fourier transform operations. Consequently, the charging process ensures a computational complexity of $\mathcal{O}(\mathcal{L}\log\mathcal{L})$ without introducing additional training overhead. For the adaptive threshold mechanism, its implementation via convolution operations is well-suited for GPU acceleration. Additionally, since $\alpha_k > 0$, a greater number of spikes in the previous timestep increases the threshold at time t, thereby promoting sparser spikes.

In the backward propagation stage, the adaptive threshold removes temporal dependencies between the gradient, enabling highly parallelizable training. Given $\mathcal{I}[t] = w^l S^{l-1}[t]$, the gradient of the loss with respect to the weight $w$ can be expressed as:

$$\nabla_{w^l}\mathcal{L} = \sum_{t=0}^{T} \underbrace{\frac{\partial\mathcal{L}}{\partial\mathcal{S}^l[t]}\frac{\partial\mathcal{S}^l[t]}{\partial\mathcal{Z}^l[t]}\frac{\partial\mathcal{Z}^l[t]}{\partial w^l}}_{\text{Sequential Training}} = \left\langle \underbrace{\frac{\partial\mathcal{L}}{\partial\mathcal{S}^l[t]}\frac{\partial\mathcal{S}^l[t]}{\partial\mathcal{Z}^l[t]}, (\mathcal{K} * \mathcal{S}^{l-1})[t]}_{\text{Parallel Training}} \right\rangle, \quad (15)$$

$\langle\cdot,\cdot\rangle$ is the inner product. The derivative $\frac{\partial\mathcal{S}^l[t]}{\partial\mathcal{Z}^l[t]}$ can be further defined as follows:

$$\frac{\partial\mathcal{S}[t]}{\partial\mathcal{Z}[t]} = C^l\mathcal{G}\left(C^l\Re\{Z[t]\} - V_{\text{th}}[t]\right)\frac{\partial\Re\{\mathcal{Z}[t]\}}{\partial\mathcal{Z}[t]}, \quad (16)$$

$\mathcal{G}(\cdot)$ denotes the surrogate gradient function. In this work, it is implemented as a double Gaussian function [72]. As shown in Eq. 15 and Eq. 16, the gradient during backpropagation depends only on the current timestep. Consequently, the incorporation of the adaptive threshold introduces no additional training complexity, while encouraging sparse spike activity and preserving low computational cost. Detailed proof is provided in the Appendix. B.

## 5 Experiment

### 5.1 Compare with the SOTA methods

To validate the effectiveness of our proposed method, we conduct experiments on multiple time-series datasets. All experiments are conducted at least five times. First, we compare the performance of D-RF with other SOTA models on commonly datasets, including Spiking Heidelberg Digits (SHD) [8] with 250 timesteps, Sequential MNIST, Permuted Sequential MNIST (S/PS-MNIST) [31] with 784 timesteps, and the more challenging Sequential CIFAR10 (S-CIFAR10) [7] with 1024 timesteps. As presented in Table 1, D-RF achieves SOTA performance while using fewer or comparable parameters.

Table 1: Performance Comparison of Various Models.

| Dataset | Method | Model Size | Type | Parallel | Dendritic | Acc. |
|---|---|---|---|---|---|---|
| S/PS-MNIST (784 Timesteps) | LIF [79] | 85.1K | FF | ✗ | ✗ | 72.06 / 10.00 |
| | ALIF [72] | 156.3K | Rec | ✗ | ✗ | 98.70 / 94.30 |
| | BRF [21] | 68.9K | Rec | ✗ | ✗ | 99.10 / 95.20 |
| | PSN [14] | 2.5M | FF | ✗ | ✓ | 97.90 / 97.80 |
| | TC-LIF [79] | 155.1K | Rec | ✗ | ✓ | 99.20 / 95.36 |
| | DH-LIF [80] | 0.8M | Rec | ✗ | ✓ | 98.9 / 94.52 |
| | PMSN [7] | 156.4K | FF | ✓ | ✓ | 99.50 / 97.80 |
| | **Ours** | 155.1K | FF | ✓ | ✓ | **99.50 / 98.20** |
| SHD (250 Timesteps) | LIF [74] | 249.0K | Rec | ✗ | ✗ | 84.00 |
| | ALIF [72] | 141.3K | Rec | ✗ | ✗ | 84.40 |
| | BRF [21] | 108.8K | Rec | ✗ | ✗ | 92.50 |
| | PSN [14] | 232.5K | FF | ✓ | ✗ | 89.75 |
| | TC-LIF [79] | 141.8K | Rec | ✗ | ✓ | 88.91 |
| | DH-LIF [80] | 0.5M | Rec | ✗ | ✓ | 91.34 |
| | PMSN [7] | 199.3K | FF | ✓ | ✓ | 95.10 |
| | **Ours** | 155.1K | FF | ✓ | ✓ | **96.20** |
| S-CIFAR10 (1024 Timesteps) | LIF [74] | 0.18M | FF | ✗ | ✗ | 45.07 |
| | PSN [14] | 6.47M | FF | ✓ | ✓ | 55.24 |
| | SPSN [14] | 0.18M | FF | ✓ | ✓ | 70.23 |
| | PMSN [7] | 0.21M | FF | ✓ | ✓ | 82.14 |
| | **Ours** | 0.21M | FF | ✓ | ✓ | **84.30** |

Additionally, experimental results demonstrate that the RF-based model consistently outperforms the similarly sized LIF model across all datasets. It further confirms the temporal modeling capabilities of RF neurons in long sequence tasks. Moreover, our model also demonstrates superior recognition performance compared to other dendritic models. Compared to the DH-LIF model [80], our approach enables parallel computation, effectively lowering the training speed associated with dendritic neurons. Compared to the PMSN model [7], our threshold resetting strategy prevents frequent spike generation and more effectively captures temporal dependencies. On the more challenging Sequential CIFAR10 dataset, our method achieves a recognition accuracy of 84.20%, representing a 2.16% improvement.

We also evaluate our approach on the more challenging LRA benchmark [60]. As shown in Table 2, our model achieves significantly higher recognition accuracy than other neural models. Notably, it

achieves 60.02% accuracy on the ListOps task. It also demonstrates strong performance on tasks with longer timesteps, reaching 86.52% precision on the Text task (4096 timesteps) and 90.02% precision on the Retrieval task (4000 timesteps). For the Image Task, D-RF attains 85.32% accuracy which underperforming SpikingSSM [51]. This gap stems from the use of LayerNorm in SpikingSSMs, which reduces temporal variance. Furthermore, the performance gap between our model and ANN-based approaches like S4 [19] is no greater than 3%. Specifically, our model's accuracy is 0.42% higher for ListOps tasks. These findings highlight the strong temporal modeling ability of our method.

Table 2: Comparison of Model Accuracy on LRA Benchmark.

| Model (Input length) | SNN | ListOps (2,048) | Text (4,096) | Retrieval (4,000) | Image (1,024) | Pathfinder (1,024) | Avg. |
|---|---|---|---|---|---|---|---|
| Random | - | 10.00 | 50.00 | 50.00 | 10.00 | 50.00 | 34.00 |
| Transformer [62] | ✗ | 36.37 | 64.27 | 57.46 | 42.44 | 71.40 | 54.39 |
| S4 (Bidirectional) [19] | ✗ | 59.60 | 86.82 | 90.90 | 88.65 | 94.20 | 84.03 |
| Binary S4D [57] | ✓ | 54.80 | 82.50 | 85.03 | 82.00 | 79.79 | 77.39 |
| → + GSU & GeLU | ✓ | 59.60 | 86.50 | 90.22 | 85.00 | 91.30 | 82.52 |
| SpikingSSMs [51] | ✓ | 60.23 | 80.41 | 88.77 | 88.21 | 93.51 | 82.23 |
| Spiking LMU [33] | ✓ | 37.30 | 65.80 | 79.76 | 55.65 | 72.68 | 62.23 |
| ELM Neuron [56] | ✓ | 44.55 | 75.40 | 84.93 | 49.62 | 71.15 | 69.25 |
| SD-TCM [24] | ✓ | 59.20 | 86.33 | 89.88 | 84.77 | 91.76 | 82.39 |
| **Ours** | ✓ | **60.02** | **86.52** | **90.02** | **85.32** | **92.36** | **82.88** |

## 5.2 Sparser Spike with Accelerated Training

To evaluate the sparsity of the D-RF method, we compare the spike firing rates and theoretical energy consumption [45] with those of similar approaches on the LRA dataset [60]. As shown in Table 3, our model demonstrates a significant advantage. Specifically, on the ListOps task [39], it achieves a Spiking Rate of 9.8% and an energy consumption of 62.48mJ. Moreover, on the Image tasks [30], the spike firing rate is reduced by 49.7% compared to the SD-TCM method. These results highlight the effectiveness of the adaptive threshold mechanism in enhancing energy efficiency.

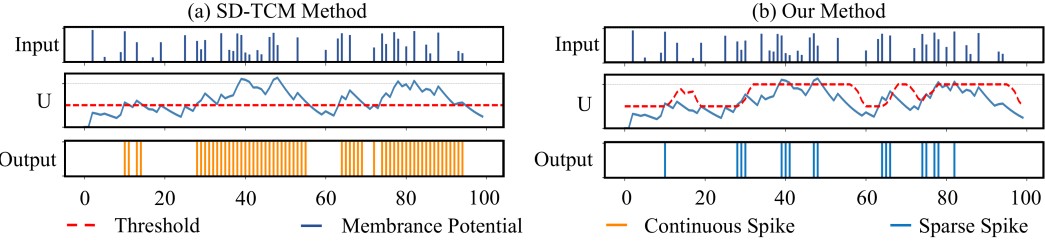

Figure 3: Comparison of Spiking Behavior Across Different Methods: (a) The SD-TCM method exhibits continuous spiking activity, as reflected in its membrane potential and spike output. (b) The D-RF method shows sparser spike generation, indicating more efficient spiking behavior.

Additionally, we visualize the spiking behaviors of different reset methods to further the sparsity of our method. The SD-TCM model [24] assumes that a learnable decay constant can effectively approximate the reset process, with a static threshold. In contrast, our method dynamically adjusts the threshold based on previous timesteps. As shown in Fig. 3, the proposed method significantly increases spike sparsity, indicating greater potential for energy efficiency.

Our method also ensures high training efficiency. We compare the per-epoch training cost across different sequence lengths. As shown in Fig. 4(a), we visualize the average epoch time under a batch size of 128. The results indicate that the proposed method achieves higher execution efficiency as the sequence length increases. At a sequence length of 32768, our method achieves a 581× speedup over BPTT [68]. Furthermore, compared to the SDN method [51], our method also shows competitive performance, achieving 4.2× acceleration. We further validate the strategy on Text tasks (4096). As

Table 3: Comparison of Metrics across the LRA Benchmark.

| Metric | Method | ListOps | Text | Retreival | Image | Pathfinder | Avg. |
|---|---|---|---|---|---|---|---|
| Spiking Rate (%) | SpikingSSM [51] | 13.2 | 10.1 | 6.9 | 22.1 | 7.4 | 11.9 |
| | SD-TCM [24]† | 11.2 | 7.9 | 5.7 | 15.7 | 5.8 | 9.3 |
| | **Ours** | **9.8** | **6.3** | **3.3** | **7.9** | **3.2** | **6.1** |
| Energy Cost (mJ) | SpikingSSM [51] | 84.2 | 355.2 | 237.0 | 708.9 | 65.1 | 290.1 |
| | SD-TCM [24]† | 71.4 | 277.8 | 195.7 | 503.6 | 51.0 | 220.6 |
| | **Ours** | **62.5** | **221.5** | **113.3** | **253.4** | **28.1** | **135.8** |

† Results reproduced by ourselves, as the original code is not publicly available.

shown in Fig. 4(b), benefiting from highly parallelized membrane potential accumulation and spike generation processes, D-RF achieves faster simulation on GPU, yielding a $1.1 \times$ speedup. Moreover, the adaptive threshold mechanism enables the transformation of the serial accumulation-decay-firing process into a parallelizable form, resulting in up to $147\times$ training acceleration. These results confirm that D-RF effectively addresses the high training cost.

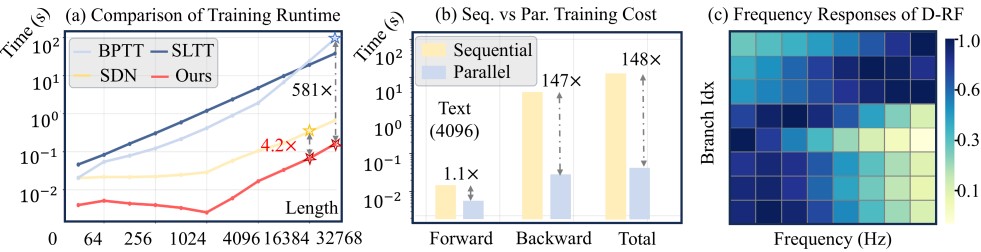

Figure 4: Train Speed and Frequency Analysis of D-RF: (a) Comparison of Training Runtime. (b) Sequential vs. Parallel Training Cost: Our method accelerates both forward and backward propagation. (c) Frequency Responses of D-RF: D-RF exhibits a broader frequency response.

## 5.3 Ablation Experiment

To assess the impact of the dendritic structure and the adaptive threshold mechanism, we conduct ablation experiments on the S-CIFAR10 and ListOps datasets. We compare the performance in different numbers of dendrites (n = 1, 4, 8, 16). As shown in Table 4, the model performance improves with an increasing number of dendrites. Considering the

Table 4: Ablation Experiment

| Dataset | Method | Number of Dendrites | | | |
|---|---|---|---|---|---|
| | | n=1 | n=4 | n=8 | n=16 |
| S-CIFAR10 | adaptive | 80.3 | 84.3 | 84.6 | **85.1** |
| | w/o | 79.2 | 83.9 | 84.1 | 84.9 |
| ListOps | adaptive | 55.2 | 59.1 | 60.2 | **60.3** |
| | w/o | 54.2 | 58.9 | 59.2 | 59.6 |

trade-off between complexity and performance, we set the number of dendrites to $n = 4$ for the S-CIFAR10 dataset and $n = 8$ for the LRA dataset. As shown in Fig. 3(c), we visualize the frequency responses of individual dendritic branches. The results indicate that the proposed D-RF neuron captures nearly the entire frequency spectrum, further confirming the effectiveness of the dendritic structure. In addition, we evaluate the effectiveness of the adaptive threshold mechanism. Experimental results demonstrate that the adaptive threshold improves model performance, primarily by effectively suppressing the adverse impact of redundant feature information on the results.

## 6 Conclusion

Inspired by the dendritic structure of biological neurons, this study proposes the D-RF model to further enhance the performance of SNNs on time-series signals. It consists of a multi-dendritic and soma structure. The multi-dendritic structure consists of branches with distinct decay factors, enabling the neuron to effectively extract multi-frequency information from the input signal. The soma with an adaptive threshold that ensures sparse spiking while enabling parallelizable computation. Extensive experiments demonstrate that our model achieves competitive results while maintaining sparse spiking activity for efficient training. These results highlight the strong potential of D-RF method to enable effective and efficient long sequence modeling on edge-computing platforms.

# 7  Acknowledgments

This work is supported in part by the National Natural Science Foundation of China (No. 62220106008 and 62271432), in part by Sichuan Province Innovative Talent Funding Project for Post-doctoral Fellows (BX202405), in part by the Shenzhen Science and Technology Program (Shenzhen Key Laboratory, Grant No. ZDSYS20230626091302006), in part by the Program for Guangdong Introducing Innovative and Entrepreneurial Teams, Grant No. 2023ZT10X044, and in part by the State Key Laboratory of Brain Cognition and Brain-inspired Intelligence Technology, Grant No. SKLBI-K2025010.

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

# A  Detail of Resonate-and-Fire Neuron

## A.1  Dynamics of Resonate-and-Fire Neuron

The RF neuron process can be modeled as a first-order linear oscillatory system:

$$\frac{d}{dt}z(t) = (b + \mathrm{i}\omega)z(t) + \mathcal{I}(t), \tag{1}$$

where $z(t) = u(t) + iv(t) \in \mathbb{C}$ represents the complex state of the neuron. $b < 0$ is the damping factor, $\omega > 0$ is the angular frequency, and $\mathcal{I}(t)$ is the external input signal to the neuron.

## A.2  Discretization via Exponential Euler Method

We use the exponential Euler method to discretize the continuous differential equation as follows:

$$z(t + \delta) = \exp\left\{(b + \mathrm{i}\omega)\delta\right\}z(t) + \int_t^{t+\delta} \exp\{(b + i\omega)(t + \delta - s)\}\mathcal{I}(s)ds, \tag{2}$$

where $\delta$ is discrete the timesteps. Assuming that the input $\mathcal{I}(s)$ remains approximately constant over the small interval $[t, t + \delta]$. This discretization can be derived as follows:

$$
\begin{aligned}
z(t + \delta) &= \exp\{(b + \mathrm{i}\omega)\delta\}z(t) + \int_t^{t+\delta} \exp\{(b + i\omega)(t + \delta - s)\}\mathcal{I}(s)ds \\
&\approx \exp\{(b + \mathrm{i}\omega)\delta\}z(t) + \mathcal{I}(t)\int_t^{t+\delta} \exp\{(b + i\omega)(t + \delta - s)\}ds \\
&= \exp\{(b + \mathrm{i}\omega)\delta\}z(t) + \mathcal{I}(t)\left[\frac{-1}{b + i\omega}\exp\{(b + i\omega)(t + \delta - s)\}\right]_{s=t}^{s=t+\delta} \\
&= \exp\{(b + \mathrm{i}\omega)\delta\}z(t) + \mathcal{I}(t)\left(\frac{-1}{b + i\omega}e^0 - \frac{-1}{b + i\omega}\exp\{(b + i\omega)\delta\}\right) \\
&= \exp\{(b + \mathrm{i}\omega)\delta\}z(t) + \mathcal{I}(t)\frac{1 - \exp\{(b + i\omega)\delta\}}{b + i\omega}.
\end{aligned}
\tag{3}
$$

When $\delta \to 0$, the second term can be represented using Taylor expansion as:

$$\mathcal{I}(t)\frac{1 - \exp\{(b + \mathrm{i}\omega)\delta\}}{b + \mathrm{i}\omega} \approx \delta\mathcal{I}(t) + \mathcal{O}(\delta^2). \tag{4}$$

Therefore, the discretized form of Eq.1 can be expressed as:

$$z[t] = \exp\{(b + \mathrm{i}\omega)\delta\}z[t - 1] + \delta\mathcal{I}[t]. \tag{5}$$

$z[t]$ represents the complex state at discrete timestep $t$, $\delta$ is the time interval, and $\mathcal{I}[t]$ is the external input at timestep $t$.

## A.3  Frequency Band Preference Characteristics

To examine the RF neuron's response to the periodic input, we seek a particular solution to Eq. 5 with the same frequency as $\mathcal{I}(t)$:

$$\mathcal{I}[t] = \mathcal{I}_0\exp\{\mathrm{i}\Omega t\delta\}, \quad z[t] = H\exp\{\mathrm{i}\Omega n\delta\}, \tag{6}$$

where $H$ is a complex constant that determines both amplitude and the phase of the response. It can be rewritten as follows:

$$H\exp\{\mathrm{i}\Omega n\delta\} = \exp\{\delta(b + \mathrm{i}\omega)\} + \delta A. \tag{7}$$

Therefore, the $H$ can be defined as follows:

$$H = \frac{\delta A}{\exp\{\mathrm{i}\Omega\delta\} - \exp\{(b + \mathrm{i}\omega)\delta\}} = \frac{\delta A}{\exp\{\mathrm{i}\Omega\delta\}\left(1 - \exp\{(b + \mathrm{i}\omega)\delta\}\exp\{-\mathrm{i}\Omega\delta\}\right)}, \tag{8}$$

The magnitude of the transfer function is:

$$\|H(\exp\{\mathrm{i}\Omega\})\| = \left\|\frac{\delta}{1 - \exp\{\delta b + \mathrm{i}(\delta\omega - \Omega)\}}\right\|. \tag{9}$$

Therefore, a single RF neuron can be regarded as a first-order band-pass filter with a resonance peak at $\Omega \approx \omega$, and a narrow frequency band determined by the damping factor $\delta b$.

For the phase diagram of RF neuron, the phase shift between input and response provides additional information about resonance properties:

$$\phi(\Omega) = \arg(H\{i\Omega\}) = -\tan^{-1}\left(\frac{\Omega - \omega}{b}\right). \tag{10}$$

At resonance $\Omega = \omega$, the phase shift is -90°. Therefore, when resonance occurs, the phase of RF neurons will accumulate rapidly.

### A.4 Reset Mechanism in Resonate-and-Fire Neuron

#### A.4.1 Traditional Soft and Hard Reset

The traditional soft and hard reset mechanisms for the RF neuron can be defined as follows:

$$\Im z[t] = \begin{cases} 0, & \text{if } \Im\{z[t]\} > V_{th}, \text{ hard reset} \\ \Im\{z[t]\} - V_{th}, & \text{if } \Im\{z[t]\} > V_{th}, \text{ soft reset}, \end{cases} \tag{11}$$

As shown in Eq. 11, when the imaginary part of the RF neuron's state exceeds the threshold, a spike is generated. For the hard reset process, the imaginary part of the RF neuron will be reset to 0. For the soft reset process, the imaginary part of the RF neuron will be subtracted from the threshold. Both of them disrupt the original oscillatory dynamics of the RF neuron.

Under reset conditions, the system becomes nonlinear, preventing direct application of the Z-transform. However, we can model the reset mechanism as a nonlinear perturbation term:

$$z_R[t] = \exp\{(b + i\omega)\} \cdot z_R[t-1] + \mathcal{I}[t] + d[t], \tag{12}$$

where $d[t]$ represents the perturbation introduced by the reset operation:

$$d[t] = \sum_k (z_R[t_k] - z[t_k]) \cdot \delta[t - t_k], \tag{13}$$

with $\delta[t - t_k]$ being the unit impulse at time $t_k$. We applying the Z-transform yields:

$$Z\{z_R[t]\} = \frac{Z\{I[t]\} + Z\{d[t]\}}{1 - \exp\{(b + i\omega)\}z^{-1}}. \tag{14}$$

For periodic reset patterns, the frequency response contains a series of harmonic components:

$$H_R(\omega') = H(\omega') + \sum_{n=-\infty}^{\infty} c_n \cdot \delta(\omega' - \omega - n\omega_r), \tag{15}$$

where $H(\omega') = \frac{1}{1 - e^{(b+i\omega)}e^{-i\omega'}}$ represents the frequency response of the original system, $c_n$ are the Fourier coefficients, and $\omega_r$ is the reset frequency. This demonstrates that the reset mechanism introduces sideband components in the frequency domain, thereby weakening the frequency selectivity of RF neurons. Therefore, conventional reset mechanisms inevitably compromise the band-selectivity property of RF neurons to some extent.

#### A.4.2 No Reset Mechanism

In the absence of a reset mechanism, the band-selectivity of RF neurons can be effectively preserved; however, it will lead to frequent spike emissions.

Consider a sinusoidal input at the resonant frequency: $\mathcal{I}[t] = \mathcal{I}_o \exp\{i\omega t\}$. The solution to the difference equation can be obtained by recursive expansion:

$$\begin{aligned} z[t] &= \exp\{(b + i\omega)t\} \cdot z[0] + \mathcal{I}_0 \sum_{k=0}^{t-1} \exp\{(b + i\omega)(t - k - 1)\} \cdot \exp\{i\omega k\} \\ &= \exp\{(b + i\omega)t\} \cdot z[0] + \mathcal{I}_0 \cdot \exp\{i\omega t\} \cdot \frac{1 - \exp\{bt\}}{1 - \exp\{b\}}. \end{aligned} \tag{16}$$

When $b < 0$ and $t >> 0$, the first term vanishes, yielding:

$$z[t] \approx \frac{\mathcal{I}_0 \exp\{i\omega t\}}{1 - \exp\{b\}}, \quad \mathrm{Im}(z[t]) \approx \frac{I_0 \sin(\omega t)}{1 - \exp\{b\}}. \tag{17}$$

Therefore, in long sequence tasks, RF neurons are more likely to exhibit frequent spike emissions. The duration of each spike is given by:

$$\Delta t_{\mathrm{spike}} = \frac{1}{\omega} \left( \pi - 2 \sin^{-1} \left( \frac{V_{th}(1 - \exp\{b\})}{I_0} \right) \right). \tag{18}$$

where $V_{th}$ is the threshold. The maximum spike duration corresponds to half of the oscillation cycle. However, in the absence of a reset mechanism, the time-invariant parameter design allows the RF neuron to be interpreted as a convolution between a fixed kernel $\mathcal{K}$ and the input $\mathcal{I}[t]$. This structure offers a clear advantage during training, as it requires only $\mathcal{O}(\mathcal{L})$ computational complexity.

### A.4.3 Adaptive Decay Factor for RF Neuron

To balance the frequency band selectivity of RF neurons and reduce their firing rates, Higuchi et al. [21] introduce a refractory mechanism into RF neurons, effectively suppressing excessive spike emissions. This mechanism consists of two main components:

$$V_{\mathrm{th}}(t) = \theta_c + q(t), \quad b(t) = b_c - q(t), \tag{19}$$

where $\theta_c$ is constant threshold and $b_c$ is the decay factor. $q(t)$ is the refractory period, which decays exponentially with time:

$$q(t) = \gamma q(t - 1) + s[t - 1]. \tag{20}$$

Here, $\gamma = 0.9$ is the default period constant. We further analyze the impact of the time-varying design of $b(t)$ in conjunction with the frequency response, showing that it does not compromise the frequency band selectivity of RF neurons. As shown in Eq. 6–9, the frequency response of the BRF neuron at time t can be defined as:

$$H(t, \omega') = \frac{1}{1 - \exp\{(b(t) + i\omega - i\omega')\}}, \tag{21}$$

This represents the system's response strength at time t to an input signal with frequency $\omega'$. We can verify that the expression attains its maximum at $\omega' = \omega$ by computing the derivative of this part. By taking the derivative of w, we can solve the extreme point situation of Eq. 21:

$$\frac{\partial |H(t, \omega')|}{\partial \omega'} = k^2 \cdot \frac{\partial}{\partial \omega'} \left[ (1 - \exp\{b(t)\} \cos(\omega - \omega'))^2 + (\exp\{b(t)\} \sin(\omega - \omega'))^2 \right]^{-\frac{1}{2}}, \tag{22}$$

where $k = |H(t, \omega')|$. For the latter term, we can further simplify.

$$\begin{aligned}
\frac{\partial |H(t, \omega')|}{\partial \omega'} &= -\frac{1}{2} k^3 \cdot \frac{\partial}{\partial \omega'} \left[ (1 - \exp\{b(t)\} \cos(\omega - \omega'))^2 + (\exp\{b(t)\} \sin(\omega - \omega'))^2 \right] \\
&= -\frac{1}{2} k^3 \cdot \frac{\partial}{\partial \omega'} \left[ 1 - 2 \exp\{b(t)\} \cos(\omega - \omega') + \exp\{2b(t)\} \right] \\
&= -\frac{1}{2} k^3 \cdot \left[ 2 \exp\{b(t)\} \sin(\omega - \omega') \right]
\end{aligned} \tag{23}$$

Therefore, for any $b$, the first-order derivative of $|H(t, \omega')|$ at $\omega' = \omega$ is zero, which is a stationary point. Therefore, we need to further derive it and find that the derivative value of the second-order derivative is:

$$\left. \frac{\partial^2 |H(t, \omega')|}{\partial \omega'^2} \right|_{\omega' = \omega} = -|H(t, \omega)|^3 \cdot e^{b(t)} < 0. \tag{24}$$

Therefore, $\omega$ is the maximum point. When $\omega' = \omega$, the band response is at its maximum value. Additionally, when a spike is emitted, the value of b(t) will decrease, resulting in a faster decay rate.

For RF neurons, the absence of a reset mechanism leads to frequent spike emissions, while simply subtracting the threshold compromises their frequency-selective properties. Although Huchigu et al. [21] preserve both energy efficiency and frequency selectivity in RF neurons, their time-invariant factor significantly hinders model training. Therefore, designing an effective and principled reset mechanism is crucial for fully exploiting the potential of RF neurons.

# B    Computational Complexity of Backpropagation in the D-RF Model

## B.1    Gradient Calculation for Backpropagation

To make the backpropagation process clearer, we begin with the gradient of the loss function with respect to the weight $w^l$, expressed as:

$$\nabla_{w^l}\mathcal{L} = \sum_{t=0}^{T} \frac{\partial\mathcal{L}}{\partial S^l[t]} \cdot \frac{\partial S^l[t]}{\partial Z^l[t]} \cdot \frac{\partial Z^l[t]}{\partial w^l}. \tag{25}$$

Since $S^l[t]$ is derived from the membrane potential $Z^l[t]$ through a nonlinear activation function, we need to compute the derivative of the spike signal with respect to the membrane potential. In this work, it is implemented as a double Gaussian function [72]. Therefore, this derivative is given by:

$$\frac{\partial S^l[t]}{\partial Z^l[t]} = C^l \mathcal{G}\left(C^l \Re\{Z[t]\} - V_{\text{th}}[t]\right) \frac{\partial\Re\{Z[t]\}}{\partial Z[t]}, \tag{26}$$

where $\mathcal{G}(\cdot)$ is the surrogate gradient function, $C^l$ is the coefficient for the current layer, and $V_{\text{th}}[t]$ is the adaptive threshold. The membrane potential $Z^l[t]$ is the weighted sum of the previous layer's spike signals $S^{l-1}[t]$. Therefore, the derivative of the membrane potential with the weight $w^l$ is:

$$\frac{\partial Z^l[t]}{\partial w^l} = \sum_{k=0}^{t} \Gamma_l \exp\{k \cdot \delta D\} \cdot S_{l-1}[t-k], \tag{27}$$

where $S_{l-1}[t-k]$ represents the spike signal from the previous layer, $\Gamma_l$ is a constant, and $\delta D$ is a parameter associated with dendritic branches. Substituting the above expressions into the gradient formula, we can obtain the following:

$$\nabla_{w^l}\mathcal{L} = \sum_{t=0}^{T} \frac{\partial\mathcal{L}}{\partial S^l[t]} \cdot C^l \mathcal{G}\left(C^l \Re\{Z[t]\} - V_{\text{th}}[t]\right) \cdot \left(\sum_{k=0}^{t} \Gamma_l \exp\{k \cdot \delta D\} \cdot S_{l-1}[t-k]\right). \tag{28}$$

For faster training, we transform the gradient computation into a parallel computation format using convolution. Thus, the final expression becomes:

$$\nabla_{w^l}\mathcal{L} = \left\langle \frac{\partial\mathcal{L}}{\partial S^l[t]} \cdot \frac{\partial S^l[t]}{\partial Z^l[t]}, (K * S^{l-1})[t] \right\rangle, \tag{29}$$

where $\langle\cdot,\cdot\rangle$ represents the inner product, and $(K * S^{l-1})[t]$ represents the convolution operation. The final gradient expression can be summarized as:

$$\nabla_{w^l}\mathcal{L} = \sum_{t=0}^{T} \frac{\partial\mathcal{L}}{\partial S^l[t]} \cdot \frac{\partial S^l[t]}{\partial Z^l[t]} \cdot \frac{\partial Z^l[t]}{\partial w^l} = \left\langle \frac{\partial\mathcal{L}}{\partial S^l[t]} \cdot \frac{\partial S^l[t]}{\partial Z^l[t]}, (K * S^{l-1})[t] \right\rangle. \tag{30}$$

This process demonstrates how the gradient calculation proceeds from the error term to the final weight update rule, incorporating both sequential and parallel training strategies for efficiency.

## B.2    Analysis of Computational Complexity

As shown in Eq. 30, our method converts gradient computation into convolution operations, effectively leveraging the efficiency of the Fast Fourier Transform (FFT). This results in a convolution computation time of $\mathcal{O}(\mathcal{L}\log\mathcal{L})$. Since the FFT of length $\mathcal{L}$ can be computed in $\mathcal{O}(\mathcal{L}\log\mathcal{L})$, and convolution is equivalent to multiplication in the frequency domain, the complexity of convolution reduces from $\mathcal{O}(\mathcal{L}^2)$ in sequential training to $\mathcal{O}(\mathcal{L}\log\mathcal{L})$ in parallel training. Therefore, using the convolution-based parallel training approach ensures that the overall computational complexity of the backpropagation process is reduced to $\mathcal{O}(\mathcal{L}\log\mathcal{L})$, making it significantly more efficient.

This analysis demonstrates that by utilizing parallel computation and FFT-based convolution, we can significantly reduce training costs, lowering the complexity from $\mathcal{O}(\mathcal{L}^2)$ to $\mathcal{O}(\mathcal{L}\log\mathcal{L})$, which is critical for long sequence modeling tasks.

## C  Experiment Detail

### C.1  Dataset Description

We provide more context and details for (P)S-MNIST and the tasks of the LRA [60]. The following descriptions primarily reference [55].

- **Sequential MNIST** (S-MNIST) [31]: Each $28 \times 28$ grayscale MNIST image is reshaped into a one-dimensional sequence of 784 scalar values. The model must assign one of ten digit labels (0–9) based solely on this temporal input.

- **Permuted Sequential MNIST** (PS-MNIST) [31]: Follows the same flattening procedure as sMNIST to produce a 784-length sequence, which is then reordered by a fixed permutation before classification into digits 0–9.

- **Spiking Heidelberg Digits** (SHD) [8]: Consists of 10 000 spike-encoded patterns representing handwritten digits (0–9) from the Heidelberg Digits dataset. Each sample is provided as either binary or continuous spike trains for classification.

- **Sequential CIFAR-10** (S-CIFAR10) [7, 14]: Converts each $32 \times 32$ color image into a sequence of 1 024 RGB triplets. The task is to classify the image into one of ten categories based on this sequential representation.

- **ListOps** [39]: A dataset for evaluating sequence-based models. It contains mathematical operations (e.g., min, max) and integer operands ranging from 0–9, represented using prefix notation and brackets. The task is to compute the result of a mathematical expression (e.g., [max 2 6 [min 9 7] 0] → 7). The characters are encoded as one-hot vectors and the sequences are padded to a maximum length of 2,000. There are 10 classes representing the results of the expression.

- **Text** [36]: A sentiment classification dataset based on the iMDB movie review dataset. The task is to classify a given movie review as positive or negative. The characters are encoded as integer tokens and the sequences are padded to a maximum length of 4,096. There are two classes representing positive and negative sentiment. The dataset includes 25,000 training samples and 25,000 test samples.

- **Retrieval** [43]: A dataset based on the ACL Anthology Network corpus. The task is to classify whether two text citations are equal or not. Each citation is encoded as a sequence of integer tokens. References is compressed separately and then passed to the final classification layer. This dataset evaluates the model's ability to represent and retrieve textual relations.

- **Image** [30]: An image classification dataset based on the CIFAR-10 dataset. It contains 32×32 grayscale images that are flattened into a sequence of length 1,024. The task is to classify each image into one of ten categories. The dataset contains 45,000 training samples, 5,000 validation samples, and 10,000 test samples.

- **Pathfinder** [32] A dataset for path finding tasks. It contains 32×32 grayscale images, each showing a start and end point represented as small circles. The task is to classify whether there is a dashed line (or path) connecting the start and end points. The sequences are padded to a length of 1,024. The dataset includes 160,000 training samples, 20,000 validation samples, and 20,000 test samples. The data were normalized to the range $[-1, 1]$.

### C.2  Model Architecture and Hyperparameter Setting

In this task, we utilize a network architecture composed of stacked residual blocks (RB) between membrane potentials. Each RB block comprises a residual connection and a sequence of "D-RF Model – 1×1 convolution – Spiking Neuron model – 1×1 convolution". Except for the possible floating-point multiplications involved in the neuron dynamics, all other operations are spike-driven, which is more favorable for deployment on resource-constrained edge devices. All experiments are conducted on Ubuntu server equipped with NVIDIA GeForce RTX 4090 (24G Memory), Intel(R) Xeon(R) Platinum 8370C CPU@2.80GHz. PyTorch 2.1.0, and CUDA 11.8.

Furthermore, we conducted a parameter count comparison between our model and baseline methods on the LRA dataset, further underscoring the efficiency of our approach. The learning rate for

Table 5: Hyperparameters for LRA Task

| Task | Depth | Norm | Channels | Pre-norm | Dropout | B | Epochs | Weight Decay |
|------|-------|------|----------|----------|---------|---|--------|--------------|
| **ListOps** | 8 | BatchNorm | 128 | False | 0 | 50 | 40 | 0.05 |
| **Text** | 6 | BatchNorm | 256 | True | 0 | 16 | 32 | 0.05 |
| **Retrieval** | 6 | BatchNorm | 256 | True | 0 | 32 | 20 | 0.05 |
| **Image** | 6 | BatchNorm | 512 | False | 0.1 | 50 | 200 | 0.05 |
| **Pathfinder** | 6 | BatchNorm | 256 | True | 0.05 | 64 | 200 | 0.03 |

neuron-specific parameters is set to 0.001, while a global learning rate of 0.005 is applied to the entire network. Detailed hyperparameter settings are summarized in Table 5.

As shown in Table 6, we report the model sizes across various LRA tasks, demonstrating that our approach maintains parameter efficiency comparable to other baselines. When combined with the performance metrics in Table 2 and the energy consumption results in Table 3. it becomes evident that our model not only achieves competitive accuracy but also induces significantly sparser spiking activity. These results underscore its potential as an effective and efficient solution for long sequence modeling on edge platforms.

Table 6: Comparison of Model Size between Competing Networks across Tasks.

| Metric | Method | S-CIFAR10 | ListOps | Text | Retreival | Image | Pathfinder |
|--------|--------|-----------|---------|------|-----------|-------|------------|
| | S4 [19] | 308K | 815K | 843K | 3.6M | 3.6M | 1.3M |
| Parm. | SpikingSSM [51] | 308K | 815K | 843K | 3.6M | 3.6M | 1.3M |
| | SD-TCM [24]† | - | 272K | 830K | 1.1M | 4.1M | 1.3M |
| | Ours | 216K | 297K | 841K | 1.1M | 3.2M | 1.3M |

† Results reproduced by ourselves, as the original code is not publicly available.

Additionally, we further report the model parameter configurations corresponding to different dendritic branches in the ablation study. While increasing the number of dendrites. leads to a larger model and generally improves recognition performance, the gains tend to saturate. Specifically, on the image classification task, the model achieves an accuracy of 86.2% with 4 dendrites, and only a marginal increase to 86.7% when the number is doubled to 8. Given the limited performance gain relative to the parameter overhead, we select n = 4 as the default configuration for the S-CIFAR10 task.

Table 7: Ablation Experiment

| Dataset | Metric | Length | Number of Dendrites | | | | | |
|---------|--------|--------|------|------|------|------|------|------|
| | | | n=1 | n=2 | n=4 | n=6 | n=8 | n=16 |
| S-CIFAR10 | Accuracy (%) | 1024 | 80.3 | 82.1 | 84.3 | 84.4 | 84.6 | 85.1 |
| | Parameter (M) | 1024 | 209K | 213K | 216K | 219K | 222K | 234K |

# D Further Discussion & Limitation

**Further Discussion**: Similar to Transformer architectures, D-RF neurons can also function as stackable computational components. They have demonstrated strong performance across various sequential benchmarks, such as LRA benchmark and S-CIFAR10 dataset. However, the sequence lengths of these datasets remain relatively limited compared with those used in current large language models (LLMs). Recent ANN-based LLMs can process input sequences ranging from 16K to 1M tokens [1, 25, 71], but such capabilities typically rely on complex architectural designs and large parameter scales, leading to substantial computational overhead. In this paper, we adapt a neural dynamics perspective, aiming to enhance memory capacity and temporal information processing.

**Limitation**: This study has two limitations. First, DRF neurons improve model performance on long sequence tasks through efficient computation. However, the additional MAC operations they introduce and their reliance on complex-valued operations may restrict their deployment on neuromorphic chips. Second, the study is primarily focused on classification tasks and does not address regression tasks, such as text generation and long sequence prediction. Therefore, future work will extend to text generation tasks and further explore implementation strategies on neuromorphic chips.

