# OpenReview forum: "Dendritic Resonate-and-Fire Neuron for Effective and Efficient Long Sequence Modeling"
_NeurIPS.cc/2025/Conference — NeurIPS 2025 poster_

### Official Review · Reviewer_zzoM · 2025-07-01

**Clarity:** 4
**Significance:** 4
**Originality:** 3
**Rating:** 5
**Confidence:** 4

**Summary:**

The authors describe the resonate and fire (RF) neuron in the context of being able to model longer term dependencies than the more conventional LIF neuron.  They propose a dendritic RF (D-RF) neuron where some of the dynamics are modelled by dendrites, allowing a single neuron to handle multiple resonances (one per dendrite).  They further propose an adaptive threshold that can be thought of as analogous to the spike frequency adaptation that is common in LIF implementations.
Experiments on basic but reasonable datasets yield good results.

**Questions:**

One assumes (perhaps wrongly) that the original RF neuron was based on observation of biology.  To what extent do the proposed modifications retain (or not) that biological plausibility?

How does the sequence length compare with current long NLP baselines?  Speech input (at ~100 Hz sample rate) or repeated sampling of static examples surely represent long inputs, but how long is long?

**Ethical Concerns:**

["NO or VERY MINOR ethics concerns only"]

**Final Justification:**

I think this is basically a good paper. The authors responded well to allay most of my concerns so I've bumped up the recommendation to accept.

**Limitations:**

No, I don't see limitations addressed explicitly in the paper.  However, the work is speculative enough to not really merit such an analysis.

**Paper Formatting Concerns:**

At least reference 23 lacks a year.

**Quality:**

3

**Strengths And Weaknesses:**

Strengths:

Paper structure is good; subject is timely and well motivated.
The RF architecture is relatively not well studied, so the general approach is quite novel.
Results are good.

Weaknesses:
The modifications to RF are in the spirit of functions that are known to work well on LIF neurons, so the novelty could be dismissed as obvious.
The datasets used are not especially "long" when compared to the things that are used for LLMs nowadays.  Then again, the LLM state of the art is well advanced whereas this is much more speculative.
It is not obvious whether the proposed architecture remains biologically plausible.

---

> ### Author Rebuttal · Authors · 2025-07-30
>
> # Weakness 1: The modifications to RF are in the spirit of functions that are known to work well on LIF neurons, so the novelty could be dismissed as obvious.
>
> The proposed adaptive threshold mechanism fundamentally differs from spike frequency adaptation [1–2] in both implementation and computational complexity, despite their conceptual similarity.
>
> **Implementation-wise**: The adaptive threshold mechanism employs causal convolution to efficiently integrate historical spikes during inference, whereas spike frequency adaptation relies on a sequential adjustment strategy, which limits its ability to parallelize.
>
> **Computational Complexity**: The adaptive threshold mechanism achieves $\mathcal{O}(L \log L)$ complexity during training, significantly more efficient than the $\mathcal{O}(L^2)$ complexity of spike frequency adaptation.
>
> Experimental results demonstrate that for sequences of length 4096, the adaptive threshold mechanism yields a **148×** improvement in training efficiency over traditional approaches.
>
> [1] Spike frequency adaptation: bridging neural models and neuromorphic applications[J]. Communications Engineering, 2024.
>
> [2] Advancing spatio-temporal processing through adaptation in spiking neural networks[J]. Nature Communications, 2025
>
> # Weakness 2: The datasets used are not especially "long" when compared to the things that are used for LLMs nowadays. The LLM state of the art is well advanced whereas this is much more speculative.
>
> As you per, current ANN-based LLMs are capable of processing sequences ranging from 16K to 1M tokens [3–4]. However, such performance gains largely rely on **complex architectural designs** and **large parameter scales**, leading to significant computational overhead. We solve the problem from the perspective of neural dynamics, with the goal of enhancing memory and temporal processing at the neuron level. This enhancement is intended to enable a better trade-off between energy efficiency and performance, particularly under smaller model scales.
>
> D-RF model provides a **viable component** for building low-power SNN-based LLMs. Future work will further investigate its modeling capability on large-scale language tasks.
>
> [3] GPT-4 Technical Report. OpenAI, 2023.
>
> [4] Qwen2.5 Technical Report. Qwen, 2024.
>
> # Weakness 3 & Question 1: One assumes (perhaps wrongly) that the original RF neuron was based on observation of biology. To what extent do the proposed modifications retain (or not) that biological plausibility?
>
> The proposed D-RF neuron is inspired by two key neurobiological observations: the **Multi-dendritic Structure** of cortical neurons and their inherent **Resonance Characteristics**.
>
> **Multi-dendritic Structure:** Neuroscientific research shown that cortical neurons are primarily composed of dendritic cells, accounting for approximately 80% of all neurons, and their complex dendritic arborizations play a crucial role in information processing [5–6].
>
> **Resonance Characteristics:** Biological studies demonstrate that resonance is an intrinsic property of neurons, characterized by subthreshold membrane potential oscillations that enable selective responses to inputs at specific frequencies [7–8].
>
> **Design of D-RF Neuron**: The D-RF neuron serves as a computational abstraction of these two biological mechanisms, aiming to maintain biological plausibility while enhancing computational efficiency. Each dendrite is modeled as a damped oscillatory dynamic, enabling efficient extraction of frequency-specific features. Additionally, the multi-dendrite and soma architecture mimics the parallel processing ability of biological neurons, allowing the model to capture multi-band information from input signals. This design achieves a favorable trade-off between biological fidelity and computational simplicity, supporting scalable deployment in large-scale neural networks.
>
> [5] Pyramidal cell types drive functionally distinct cortical activity patterns during decision-making[J]. Nature neuroscience, 2023.
>
> [6] Brain-wide presynaptic networks of functionally distinct cortical neurons[J]. Nature, 2025.
>
> [7] Oscillations emerging from noise-driven steady state in networks with electrical synapses and subthreshold resonance[J]. Nature communications, 2014
>
> [8] Resonance with subthreshold oscillatory drive organizes activity and optimizes learning in neural networks[J]. PNAS, 2023.
>
> # Question 2: How does the sequence length compare with current long NLP baselines? Speech input (at ~100 Hz sample rate) or repeated sampling of static examples surely represent long inputs, but how long is long?
>
> **Compare with long NLP baseline**: We surveyed commonly used text benchmarks, including GLUE, WikiText-103, and WikiText-2. A summary is provided in the table below:
>
> |Benchmark|Tokens|
> |:-:|:-:|
> |GLUE|10~60|
> |Wikitext103|1024~2048|
> |Wikitext2|512~1024|
> |Ours|1024~4096|
>
> Therefore, the sequence length used in this work is comparable to that of existing NLP benchmarks, indicating its potential for effective language modeling.
>
> **how long is long**: Due to the inherent training challenges of SNNs—such as gradient approximation and the instability introduced by BPTT—most existing models adopt **4** to **8** timesteps to maintain a balance between performance and training stability. Within this constraint, tasks like S/PS-MNIST (784 steps) and S-CIFAR10 (1024 steps) are commonly regarded as long-sequence benchmarks. In this context, the proposed D-RF model further demonstrates its capability to model extended sequences, achieving effective performance on the **LRA** benchmark with 1K–4K time steps.
>
> # Paper Formatting Concerns
> Thank you for the correction. Reference [23] will be revised accordingly.
>
> [23] Higuchi S, Kairat S, Bohté S M, et al. Balanced resonate-and-fire neurons[C]//Proceedings of the 41st International Conference on Machine Learning. 2024: 18305-18323.

---

> > ### Comment · Reviewer_zzoM · 2025-08-04
> >
> > Thank you for clarifications and answers. They offset most of my worries.  However, are you able to propagate these into the manuscript?

---

> > > ### Author Response · Authors · 2025-08-04
> > >
> > > We are pleased to know that most of your concerns has been addressed. Your thoughtful comments are valuable for improving the quality of the manuscript. All suggested revisions will be incorporated into the revised version to further enhance its clarity and presentation.

---

### Official Review · Reviewer_m2wP · 2025-07-02

**Clarity:** 3
**Significance:** 3
**Originality:** 4
**Rating:** 5
**Confidence:** 5

**Summary:**

This paper proposes a D-RF neuron to address efficiency and performance challenges in long sequence modeling. The model comprises multi-dendritic branches and a soma structure, where each dendritic branch leverages RF neuron oscillatory properties to capture frequency-specific signals, while the soma employs an adaptive threshold mechanism that dynamically adjusts thresholds based on historical spiking activity. Experimental results demonstrate that D-RF achieves competitive performance across multiple long sequence tasks while maintaining sparse firing and training efficiency.

**Questions:**

1. Does introducing a multi-dendritic structure into LIF neurons enhance their temporal modeling capabilities? The authors are required to conduct experiments to verify this.
2. D-RF neurons seem to introduce complex number operations. Will this affect their deployment in neuromorphic chips?
3. How to train the RF neuron-based neural networks? The authors should clarify this question.

**Ethical Concerns:**

["NO or VERY MINOR ethics concerns only"]

**Limitations:**

Yes.

**Paper Formatting Concerns:**

None.

**Quality:**

3

**Strengths And Weaknesses:**

Strengths:
1. The paper provides a systematic analysis of RF neuron limitations, identifying constrained frequency response bandwidth and energy-efficiency trade-offs as key bottlenecks.
2. The multi-dendritic architecture effectively expands frequency response capabilities, while the adaptive threshold mechanism facilitates efficient optimization.
3. The adaptive threshold design maintains O(L log L) computational complexity, enabling practical SNN training for extended sequence modeling.
4. Extensive experiments thoroughly validate the proposed method's effectiveness and efficiency.

Weaknesses:
1. Insufficient analysis of computational overhead analysis comparing D-RF to conventional spiking neurons.
2. D-RF introduces additional complex number operations, and the authors did not discuss whether this feature hinders the deployment of neuromorphic chips.
3. The selection of dendritic branch number lacks systematic analysis of the performance-complexity trade-off.

---

> ### Author Rebuttal · Authors · 2025-07-30
>
> We sincerely thank the reviewer for the valuable comments. In response to the concerns raised, we provide the following detailed replies and additional experiments:
> # Weakness 1: Insufficient analysis of computational overhead analysis comparing D-RF to conventional spiking neurons.
> We compared the computational complexity of the D-RF neuron and the LIF neuron.
>
> **Dynamics of the LIF Neuron**:
>
> $H[t] = \tau U[t-1] + \frac{1}{\tau} (I[t] - (U[t-1] - V_reset)),$
>
> $S[t] = \Theta(U[t] - V_{th}),$
>
> $U[t] = H[t] (1 - S[t]) + V_reset S[t],$
>
> where $I[t]$ denotes the input current, $V_{\text{reset}}[t]$ represents the reset membrane potential, and $\tau$ is the time constant. For a sequence of length $T$, the computational complexity of the LIF neuron is $\mathcal{O}(T)$.
>
> **Dynamics of the D-RF Neuron**:
>
> $\mathcal{Z}[t] = \exp \{\delta \mathcal{D}\} Z[t-1] + \Gamma^l \mathcal{I}[t], \quad \mathcal{D} \in \mathbb{C}^{N\times N}, \Gamma^l \in \mathbb{R},^N$
>
> $\mathcal{S}[t] = \Theta (\mathcal{C} \Re \{ Z[t]\} - \sum_{k=1}^T \alpha_k \Theta \left(\Re\{\mathcal{Z}[t-k-1]\} - V_{pre}) - V_{pre}\right), \quad \mathcal{C} \in \mathbb{R}^N,$
>
> $N$ denotes the number of dendritic branches. Therefore, the computational complexity of the D-RF neuron is $\mathcal{O}(NT)$.
>
> #  Weakness 2 and Question 2: The impact of D-RF's complex number operations on neuromorphic chip deployment remains inadequately discussed by the authors.
>
> As you per, D-RF model introduces additional complex-valued computations. It is worth noting that modern neuromorphic chips already support such operations, such as Loihi2 [1]. This provides hardware-level support for the deployment of the D-RF model.
>
> [1] Efficient neuromorphic signal processing with loihi 2[C]. SiPS 2021.
>
> # Weakness 3: The selection of dendritic branch number lacks systematic analysis of the performance-complexity trade-off.
>
> To further address your concern, we conducted ablation experiment on the S-CIFAR10 dataset:
>
> |Dataset|Metric|n=2|n=4|n=8|n=16|
> |:-:|:-:|:-:|:-:|:-:|:-:|
> |S-CIFAR10|Acc|82.1%|84.3%|84.6%|85.1%|
> |S-CIFAR10|Para|209K|213K|216K|219K|222K|234K|
>
> The model reaches 84.3% accuracy at $n=4$, with only 3K more parameters than $n=2$ and a 2.2% accuracy gain. Increasing to $n=8$ brings just a 0.3% improvement over $n=4$. Thus, $n=4$ offers the best trade-off between performance and complexity on the S-CIFAR10 tasks.
>
> # Question 1: Does introducing a multi-dendritic structure into LIF neurons enhance their temporal modeling capabilities? The authors are required to conduct experiments to verify this.
> As you noted, multi-dendrite modeling significantly enhances the temporal processing capability of neurons. To assess this effect, we replace the D-RF neuron with the LIF neuron and evaluate performance on the S-CIFAR10 task:
>
> |Method|Para|Acc|
> |:-:|:-:|:-:|
> |LIF|204k|45.07|
> |LIF with Dendritc|215k|68.10|
> |D-RF (Ours)|216k|84.30|
>
> Experimental results show that incorporating a multi-dendrite structure significantly improves the modeling capability of the LIF neuron. Based on this, the D-RF neuron achieves an additional **16.2%** performance gain, further highlighting the advantage of RF neurons in temporal processing tasks.
>
> # Question 3: How to train the RF neuron-based neural networks? The authors should clarify this question.
>
> In this work, we adapt the STBP training method [2] to effectively address the issue of non-differentiable gradients. The weight gradient is defined as follows:
> $$\frac{\partial L}{\partial W } = \sum_t \frac{\partial L}{\partial S[t]} \frac{\partial S[t]}{\partial V[t]} \frac{\partial V[t]}{\partial W},$$
> $\frac{\partial S[t]}{\partial V[t]}$ denotes the surrogate gradient of the Heaviside function. In this work, we use the arctangent function [3] as the surrogate gradient.
> $$\frac{\partial S[t]}{\partial V[t]} = \frac{\alpha}{2(1+(\frac{\pi}{2}\alpha x)^2))}.$$
>
> [2] Spatio-temporal backpropagation for training high-performance spiking neural networks[J]. Frontiers in neuroscience, 2018.
>
> [3] Temporal efficient training of spiking neural network via gradient re-weighting[C]. ICLR, 2022.

---

> > ### Comment · Area_Chair_XSsd · 2025-08-05
> >
> > Dear reviewer,
> >
> > Please read the authors' rebuttal if you haven't done so and state your response accordingly.
> >
> > Best,
> > AC

---

> > ### Comment · Reviewer_m2wP · 2025-08-07
> >
> > Thank you for your response. It  addresses my concerns regarding computational cost and temporal modeling capability. I suggest adding the relevant comparative experiments to the paper for completeness.

---

> > > ### Author Response · Authors · 2025-08-07
> > >
> > > We sincerely appreciate your response and thoughtful feedback. We are glad to know that your concerns regarding computational cost and temporal modeling capability have been adequately addressed. We will include the relevant content in the revised manuscript to further enhance its clarity and overall presentation.

---

### Official Review · Reviewer_kTeX · 2025-07-02

**Clarity:** 1
**Significance:** 3
**Originality:** 2
**Rating:** 4
**Confidence:** 4

**Summary:**

The authors propose the Dendrite Resonate-and-Fire (DRF) neuron as a novel approach to address the limitations of conventional Resonate-and-Fire (RF) neurons. To overcome the limited bandwidth response of a single RF neuron, the DRF architecture integrates dendritic branches with varying eigen-frequencies to capture complex frequency compositions. Additionally, the model incorporates a dynamic adaptive threshold mechanism with parallelization to balance training efficiency and spike sparsity. Empirical evaluations demonstrate that the DRF neuron achieves consistently improved performance across multiple time-series datasets, while maintaining comparable model size, preserving spike sparsity, and enhancing training efficiency.

**Questions:**

1. How does the summation of dendritic signals at the soma contribute to frequency selectivity, given that the signals are merged before thresholding? How are multiple selected frequencies still reflected in the output spikes?

2. What are the frequency initialization for the 8 dendrites that are shown in Figure 4 (c)? How is it possible to obtain enough information about the input signal with only 4-8 eigen-frequencies?

3. What is the memory complexity of the D-RF network?

**Ethical Concerns:**

["NO or VERY MINOR ethics concerns only"]

**Limitations:**

They have not addressed the memory usage required for parallel processing, which may be high since the whole sequence of the data is input simultaneously.

**Paper Formatting Concerns:**

No formatting concerns

**Quality:**

3

**Strengths And Weaknesses:**

Strength
- Comparison of the time series dataset extensive and shows SOTA performance in all cases.
- Faster training runtime and training energy cost efficient. Supports long sequence tasks.
- Higher spike sparsity compared to other large spiking models on the LRA benchmark.

Weakness
- PathX is missing from the LRA Benchmark
- The eigen-frequencies of the dendrites are fixed to the freq. bins of the input FFTs; in other words, depends on the discrete time delta. Would be interesting if these frequencies could be learned as well.
- Neither hyperparameters nor the architecture of the models trained are mentioned in detail on the appendix or in the supplementary material; especially for S/PS-MNIST and SHD.
- (maybe too small a point to mention) Inconsistent number of parameters for S-CIFAR10 on Tables 7 and 8.

---

> ### Author Rebuttal · Authors · 2025-07-30
>
> Thank you for your valuable suggestion. To address your concern, we have conducted additional experiments.
> # W1: PathX is missing from the LRA Benchmark
>
> Some existing SNN-based sequence models, such as Spiking LMUFormer [1] and PRF [2], encounter training difficulties or fail to converge on the Path-X task. To ensure a fair comparison, we **only** report results on the commonly tasks. Nevertheless, to address your concern, we conduct additional evaluations on the Path-X task:
>
> |Method|Acc(%)|
> |:-:|:-:|
> |Spiking LMUFormer [1]|Fail|
> |PRF [2]|Fail|
> |Binary S4D [3]|61.2%|
> |Ours|**82.8%**|
>
> [1] Lmuformer: Low complexity yet powerful spiking model with legendre memory units[C]. ICLR 2024.
>
> [2] PRF: Parallel Resonate and Fire Neuron for Long Sequence Learning in Spiking Neural Networks.
>
> [3] Learning long sequences in spiking neural networks[J]. Scientific Reports, 2024.
>
> # W2: The eigen-frequencies of the dendrites are fixed to the freq. bins of the input FFTs; in other words, depends on the discrete time delta. Would be interesting if these frequencies could be learned as well.
>
> It is important to clarify that both the frequency bands and discrete time parameters of the D-RF neuron are **learnable**. Specifically, for the D-RF neuron:
>
> $$\mathcal{Z}[t] = \exp \{\delta \mathcal{D}\} Z[t-1] + \Gamma^l \mathcal{I}[t], \quad \mathcal{D} \in \mathbb{C}^{N\times N}, \Gamma^l \in \mathbb{R}^N,$$
>
> $$\mathcal{S}[t] = \Theta (\mathcal{C} \Re \{ Z[t]\} - \sum_{k=1}^T \alpha_k \Theta \left(\Re\{\mathcal{Z}[t-k-1]\} - V_{pre}) - V_{pre}\right), \quad \mathcal{C} \in \mathbb{R}^N,$$
> where $\delta$ and $\mathcal{D}$ are **learnable** parameters, enabling the D-RF neuron to more effectively capture information from the input signal.
>
> # W3: Neither hyperparameters nor the architecture of the models trained are mentioned in detail on the appendix or in the supplementary material; especially for S/PS-MNIST and SHD.
>
> We adopt a residual block-based network architecture, where each block consists of: "D-RF Module – 1×1 Convolution – Spiking Neuron – 1×1 Convolution." The detailed configurations for each task are as follows:
>
> |Tasks|S/PS-MNIST|SHD|S-CIFAR10|ListOps|Text|Retrieval|Image|Pathfinder|PathX|
> |:-:|:-:|:-:|:-:|:-:|:-:|:-:|:-:|:-:|:-:|
> |Depth|3|3|8|6|6|6|6|6|6|
> |Channel|128|128|256|128|256|256|512|256|256|
> |Para|155.1K|155.1K|216K|297K|841K|1.1M|3.2M|1.3M|846K|
> |Acc|99.50 / 98.20|96.20|84.30|60.02|86.52|90.02|85.32|92.36|82.8%|
>
> # W4: (maybe too small a point to mention) Inconsistent number of parameters for S-CIFAR10 on Tables 7 and 8.
> Thank you for your correction. After recalculation, the parameter count for the S-CIFAR10 task is **216K**. We will further revise the manuscript accordingly.
>
> # Q1: How does the summation of dendritic signals at the soma contribute to frequency selectivity, given that the signals are merged before thresholding? How are multiple selected frequencies still reflected in the output spikes?
>
> For the D-RF neuron, its dynamics can be expressed as follows:
>
> $$\mathcal{Z}[t] = \exp \{\delta \mathcal{D}\} Z[t-1] + \Gamma^l \mathcal{I}[t], \quad \mathcal{D} \in \mathbb{C}^{N\times N}, \Gamma^l \in \mathbb{R}^N,$$
> $$\mathcal{S}[t] = \Theta (\mathcal{C} \Re \{ Z[t]\} - \sum_{k=1}^T \alpha_k \Theta \left(\Re\{\mathcal{Z}[t-k-1]\} - V_{pre}) - V_{pre}\right), \quad \mathcal{C} \in \mathbb{R}^N.$$
> The learnable dendritic parameters $\mathcal{D}$ and $\delta$ determine the neuron's response to specific frequency bands, enabling **frequency-selective filtering**. The parameter $\mathcal{C}$ adjusts the relative weighting of each frequency band. The soma integrates the weighted dendritic inputs and generates sparse spikes through an adaptive threshold mechanism. During training, all parameters are optimized via gradient-based methods to achieve task-specific frequency selection.
>
> # Question 2: What are the frequency initialization for the 8 dendrites that are shown in Figure 4 (c)? How is it possible to obtain enough information about the input signal with only 4-8 eigen-frequencies?
>
> The eight dendrites are initialized using a linear distribution. For the $k$-th input, the initial frequency is defined as follows:
> $$\omega_k = \frac{k}{8}\pi, k \in \[0,1,2,\cdots, k-1\],$$
> Through the **learnable** frequency parameters, the model adaptively adjusts to an optimal frequency-band distribution. As shown in Fig. 4c, the trained dendrites nearly cover the entire frequency spectrum, ensuring comprehensive extraction of frequency-domain information. The performance analysis for different numbers of dendrites is presented below:
>
> |Dataset|Metric|n=1|n=4|n=8|n=16|
> |:-:|:-:|:-:|:-:|:-:|:-:|
> |S-CIFAR10|Acc|80.3%|84.3%|84.6%|85.1%|
> |S-CIFAR10|Para|209k|216k|222k|234k|
> |ListOps|Acc|55.2%|59.1%|60.2%|60.3%|
> |ListOps|Para|275k|280k|288k|310K|
>
> The results show that when $n=4$ or $n=8$, the D-RF model achieves an effective balance between energy efficiency and parameter count.
>
> # Question 3: What is the memory complexity of the D-RF network?
> To further illustrate the memory requirements of the D-RF model, we provide an analysis of its memory usage and computational complexity:
>
> |Model|Period|Memory Complexity|Time Complexity|
> |:-:|:-:|:-:|:-:|
> |LIF|Train|O(1)|O(L^2)|
> |LIF|Inference|O(1)|O(L)|
> |D-RF (Ours) |Train|O(NL)|O(LlogL)|
> |D-RF (Ours) |Inference|O(N)|O(L)|
>
> Here, $N$ denotes the number of dendrites and $L$ the sequence length. As shown in the table, during training, the D-RF model leverages parallel computation to accelerate long-sequence processing. During inference, its space complexity is only $\mathcal{O}(N)$, introducing minimal additional memory overhead compared to LIF neurons.

---

> > ### Comment · Area_Chair_XSsd · 2025-08-05
> >
> > Dear reviewer,
> >
> > Please read the authors' rebuttal if you haven't done so and state your response accordingly.
> >
> > Best,
> > AC

---

> ### Comment · Reviewer_kTeX · 2025-08-07
> **Questions addressed**
>
> Thank you for your clarifications, they address most of my questions and concerns. However, I feel that Question 2 has not been fully answered on a theoretical level, though I appreciate the empirical results.
> Overall, I still consider the paper a valid contribution. That said, I increasingly feel that, methodologically, it represents only a modest increment over the PRF work. The main difference appears to be the structural composition that merges RF and ALIF units.

---

> > ### Author Response · Authors · 2025-08-07
> >
> > We are pleased to have addressed the majority of your concerns. To further clarify the remaining points, we offer the following **theoretical insights** and **design explanations**:
> >
> > # Theoretical Analysis
> >
> > As you correctly noted, theoretical analysis can further clarify the significance of incorporating a multi-dendritic structure in the D-RF neuron. We provide a concise derivation from the perspective of **frequency response**:
> >
> > $||H_\\text{D-RF}(\exp(\text{i}\Omega))|| = \sum_{i=1}^{n} \mathcal{C}_i ||\frac{\delta}{1 - \exp(\delta b + \text{i}(\delta \omega - \Omega))}||,$
> >
> > $B_\text{eff} \approx \sum_{i=1}^{n} \beta_{i} ||\frac{\tau_i}{\delta}||,$
> >
> > $n$ denotes the number of dendritic branches, and $\beta_i \in [0, 1)$ quantifies the independent contribution of the i-th branch to the overall frequency coverage. As $n$ increases, D-RF demonstrates **a broader frequency bandwidth**, thereby improving its capacity to capture and process information across a wider range of frequencies.
> >
> > Moreover, **experimental results** further demonstrate the effectiveness of the multi-dendritic structure. On the S-CIFAR10 dataset, the accuracy improves from 80.3% when $n=1$ to 85.1% when $n=16$. These findings provide strong empirical support for the efficacy and design rationale of the multi-dendritic structure.
> >
> > # Design of D-RF
> > Thank you again for your valuable feedback. As you have correctly observed, the D-RF neuron combines the architectural strengths of both RF and ALIF neurons. However, it is **not** a simple structural fusion. Rather, D-RF is the result of a carefully designed architecture that preserves the **high training efficiency** of RF neurons while maintaining the **essential sparse spiking behavior** of ALIF neurons. To validate this design, we conduct experiments on the SHD dataset:
> >
> > |Dataset|Method|Complexity|Fire Ratio|Acc (%)|
> > |:-:|:-:|:-:|:-:|:-:|
> > |SHD|ALIF [1]|$O(L^2)$|9.1%|84.4%|
> > |SHD|PRF [2]|$O(LlogL)$|11.2%|92.6%|
> > |SHD|D-RF (Ours)|$O(LlogL)$|9.4%|96.2%|
> >
> > As shown in the table, our method achieves a computational complexity comparable to PRF while effectively preserving the **sparse spiking behavior characteristic** of ALIF.
> >
> > [1] Accurate and efficient time-domain classification with adaptive spiking recurrent neural networks. Nature Machine Intelligence, 2021.
> >
> > [2] PRF: Parallel Resonate and Fire Neuron for Long Sequence Learning in Spiking Neural Networks.

---

### Official Review · Reviewer_2jgg · 2025-07-03

**Clarity:** 3
**Significance:** 4
**Originality:** 3
**Rating:** 5
**Confidence:** 4

**Summary:**

This paper proposes D-RF neurons to address the limitations of RF neurons in long sequence modeling. The work provides comprehensive theoretical analysis, revealing RF neurons' limited frequency band perception and energy-performance trade-offs through frequency domain analysis. The method is well-designed with D-RF employing multi-dendritic architecture for full-spectrum coverage and adaptive threshold mechanisms that balance training efficiency with sparse spike generation.

**Questions:**

1.To validate D-RF neurons' temporal processing capabilities, evaluation on speech datasets and neuromorphic datasets would be necessary.
2. Given the complex neural dynamics of D-RF neurons, providing pseudocode or algorithmic flowcharts would be beneficial for understanding and reproducibility.
3. The authors are encouraged to provide additional analysis of model complexity and performance variations as dendritic branches increase, to further justify the rationality of dendritic branch selection.

**Ethical Concerns:**

["NO or VERY MINOR ethics concerns only"]

**Final Justification:**

The rebuttal successfully addressed my concerns. I'll maintain my score and recommend this paper for acceptance.

**Limitations:**

Yes.

**Paper Formatting Concerns:**

No.

**Quality:**

3

**Strengths And Weaknesses:**

Strengths:
1. Comprehensive theoretical foundation: The method analyzes RF neurons' limited frequency band response and the challenge of balancing energy efficiency with training efficiency.
2. Efficient Neural Dynamics: D-RF neurons achieve full-frequency coverage through multi-dendritic structures while ensuring parallel training and sparse spike generation.
3. Competitive Performance: The method performs well on datasets of varying lengths, demonstrating efficient temporal modeling capabilities.
4. Trade-offs between efficient and train efficiency: Ablation studies demonstrate D-RF's effective balance between energy efficiency and training efficiency.

Weakness:
1. Compared to conventional spiking neurons [1, 2], D-RF neurons exhibit complex neural dynamics that introduce additional computational overhead.
2. Although the paper validates the method on mainstream sequence tasks, it lacks performance evaluation on speech datasets, neuromorphic datasets, and other relevant domains.

[1] Spiking neuron models: Single neurons, populations, plasticity[M]. Cambridge university press.
[2] Accurate and efficient time-domain classification with adaptive spiking recurrent neural networks[J]. Nature Machine Intelligence.

---

> ### Author Rebuttal · Authors · 2025-07-30
>
> Thank you very much for your valuable comments. We will further clarify and supplement the explanation to address your concerns.
> # W1: The Computational Overhead of D-RF Neuron.
> Answer: As you per, D-RF neuron exhibits more complex dynamics, which significantly enhance the temporal processing capability of SNNs. On the S-CIFAR10 dataset, our method achieves an accuracy of **84.3%**, representing a **39.23%** improvement over the 45.07% achieved by the LIF neuron. Notably, current neuromorphic chips such as Loihi2 [1] supports complex-valued computations.
>
> # W2 & Q1: Evaluation on Speech and Neuromorphic Datasets.
> Answer: To evaluate the generalizability of the proposed method, we further conducted experiments on the **Speech Dataset** (GSC) and the **neuromorphic dataset** (CIFAR10-DVS). The results are presented as follows:
> | Dataset | Method | Para | Acc(%)|
> |-|-|-|-|
> |GSC|LIF|106k|75.20|
> |GSC|**Ours**|123k|95.20|
> |CIFAR10DVS|PLIF|354k|74.80|
> |CIFAR10DVS|**Ours**|386k|86.20|
>
> # Q2: Algorithmic of D-RF Neuron.
> To facilitate understanding and reproducibility, we provide both the **parallel training and serial inference** algorithms of the D-RF neuron.
>
> |Algorithm 1: Parallel Training of D-RF Neuron|
> |-|
> |1: **Input:** $\mathcal{I}[t] \in R$|
> |2: **Output:** $\mathcal{S}[t] \in R$|
> |3: $\quad \mathcal{Z}[t] = \exp\{\delta \mathcal{D}\} \mathcal{Z}[t-1] + \Gamma^l \mathcal{I}[t]$|
> |4: $\quad V_{th}[t] = V_{pre} + \sum_{k=1}^T \alpha_k \Theta \left(\Re\{\mathcal{Z}[t-k-1]\} - V_{pre}\right)$|
> |5: $\quad \mathcal{S}[t] = \Theta(\mathcal{C}\Re\{Z[t]\} - V_{th}[t])$|
>
> |Algorithm 2: Serial Inference of D-RF Neuron|
> |-|
> |1: **Input:** $\mathcal{I} \in R^T, \mathcal{K} \in R^T$|
> |2: **Output:** $\mathcal{S} \in \{0, 1\}^T$|
> |3: $\quad \mathcal{Z} =\mathcal{F}^{-1} \{\mathcal{F}\{K\} \cdot \mathcal{F} \{I\}\} $|
> |4: $\quad V_{th} = \text{Conv1d}\{\Theta(\mathcal{C}\Re\{\mathcal{Z}\} - V_{pre})\}+V_{pre}$|
> |5: $\quad \mathcal{S} = \Theta(\mathcal{C}\Re\{Z[t]\} - V_{th})$|
>
> # Q3: Analysis of Model Complexity and Performance.
> To further justify the choice of dendritic branch number, we analyze both the **computational complexity** and **model size** of the D-RF neuron.
> **Computational Complexity**: Dynamics of the D-RF neuron is as follows:
>
> $$\mathcal{Z}[t] = \exp \{\delta \mathcal{D}\} Z[t-1] + \Gamma^l \mathcal{I}[t], \quad \mathcal{D} \in \mathbb{C}^{N\times N}, \Gamma^l \in \mathbb{R}^N,$$
> $$\mathcal{S}[t] = \Theta (\mathcal{C} \Re \{ Z[t]\} - \sum_{k=1}^T \alpha_k \Theta \left(\Re\{\mathcal{Z}[t-k-1]\} - V_{pre}) - V_{pre}\right), \quad \mathcal{C} \in \mathbb{R}^N.$$
>
> Therefore, the computational complexity of the D-RF neuron is $\mathcal{O}(NL)$, where $N$ denotes the number of dendrites and $L$ is the sequence length. The complexity increases linearly with the number of dendrites.
>
> **Model Size**: We evaluated the model’s accuracy on the S-CIFAR10 task under different numbers of dendrites.
>
> |Dataset|Metric|n=2|n=4|n=8|n=16|
> |:-:|:-:|:-:|:-:|:-:|:-:|
> |S-CIFAR10|Acc|82.1%|84.3%|84.6%|85.1%|
> ||Para|209K|213K|216K|219K|222K|234K|
>
> As shown in the table, the D-RF model achieves an accuracy of **84.3%** with 216K parameters when the number of dendrites is set to 4. Compared to $n=1$, this setting adds only 3K parameters while providing a 2.2% improvement in accuracy. In contrast, increasing to $n=8$ yields **only** a 0.3% gain in energy efficiency. Therefore, we choose $n=4$ as the optimal dendrite number for the S-CIFAR10 task, offering the **best trade-off** between model complexity and accuracy.

---

> > ### Comment · Reviewer_2jgg · 2025-08-04
> >
> > The rebuttal successfully addressed my concerns. I'll maintain my score and recommend this paper for acceptance.

---

> > > ### Author Response · Authors · 2025-08-07
> > >
> > > Thank you for your response and kind recognition. We are glad to know that your concerns have been effectively addressed. Your feedback is highly valued and plays an important role in enhancing the quality of our work.

---

### Decision · Program_Chairs · 2025-09-17

**Decision:**

Accept (poster)

**Comment:**

During the rebuttal, the authors addressed many of the reviewers' concerns and the reviewers have reached a collective positive grade. I am hence recommending an acceptance.